# Predicting Long-Term Childhood Survival of Newborns with Congenital Heart Defects: A Population-Based, Prospective Cohort Study (EPICARD)

**DOI:** 10.3390/jcm13061623

**Published:** 2024-03-12

**Authors:** Makan Rahshenas, Nathalie Lelong, Damien Bonnet, Lucile Houyel, Babak Choodari-Oskooei, Mithat Gonen, Francois Goffinet, Babak Khoshnood

**Affiliations:** 1Centre of Research in Epidemiology and Statistics (Inserm 1153, CRESS), Université Paris Cité, 75006 Paris, France; rahshenas@gmail.com (M.R.); nathalie.lelong@inserm.fr (N.L.); francois.goffinet@aphp.fr (F.G.); 2M3C-Necker, National Reference Center for Complex Congenital Heart Diseases, APHP, Université Paris Cité, Hôpital Necker-Enfants Malades, 75015 Paris, France; damien.bonnet@aphp.fr (D.B.); lucile.houyel@aphp.fr (L.H.); 3MRC Clinical Trials Unit at UCL, Institute of Clinical Trials and Methodology, University College London, London WC1E 6BT, UK; b.choodari-oskooei@ucl.ac.uk; 4Memorial Sloan-Kettering Cancer Center, New York, NY 10065, USA; gonenm@mskcc.org

**Keywords:** congenital anomalies, outcomes, prediction, epidemiology, congenital heart defects

## Abstract

**Backgroud:** Congenital heart defects (CHDs) are the most frequent group of major congenital anomalies, accounting for almost 1% of all births. They comprise a very heterogeneous group of birth defects in terms of their severity, clinical management, epidemiology, and embryologic origins. Taking this heterogeneity into account is an important imperative to provide reliable prognostic information to patients and their caregivers, as well as to compare results between centers or to assess alternative diagnostic and treatment strategies. The Anatomic and Clinical Classification of CHD (ACC-CHD) aims to facilitate both the CHD coding process and data analysis in clinical and epidemiological studies. The objectives of the study were to (1) Describe the long-term childhood survival of newborns with CHD, and (2) Develop and validate predictive models of infant mortality based on the ACC-CHD. **Methods:** This study wasbased on data from a population-based, prospective cohort study: Epidemiological Study of Children with Congenital Heart Defects (EPICARD). The final study population comprised 1881 newborns with CHDs after excluding cases that were associated with chromosomal and other anomalies. Statistical analysis included non-parametric survival analysis and flexible parametric survival models. The predictive performance of models was assessed by Harrell’s C index and the Royston–Sauerbrei $R_{D}^{2}$, with internal validation by bootstrap. **Results:** The overall 8-year survival rate for newborns with isolated CHDs was 0.96 [0.93–0.95]. There was a substantial difference between the survival rate of the categories of ACC-CHD. The highest and lowest 8-year survival rates were 0.995 [0.989–0.997] and 0.34 [0.21–0.50] for “interatrial communication abnormalities and ventricular septal defects” and “functionally univentricular heart”, respectively. Model discrimination, as measured by Harrell’s C, was 87% and 89% for the model with ACC-CHD alone and the full model, which included other known predictors of infant mortality, respectively. The predictive performance, as measured by $R_{D}^{2}$, was 45% and 50% for the ACC-CHD alone and the full model. These measures were essentially the same after internal validation by bootstrap. **Conclusions:** The ACC-CHD classification provided the basis of a highly discriminant survival model with good predictive ability for the 8-year survival of newborns with CHDs. Prediction of individual outcomes remains an important clinical and statistical challenge.

## 1. Introduction

Congenital heart defects (CHDs) comprise a heterogeneous group of structural anomalies that may affect various aspects of the normal cardiac anatomy or function; they can be “isolated” defects (i.e., involving only one or more cardiac structures and/or the adjoining vessels) or be associated with chromosomal (e.g., Down syndrome) or genetic abnormalities or those of other systems (e.g., the digestive system). CHDs may also comprise one of the elements of a known syndrome (e.g., the Di George syndrome) [1,2,3,4].

The signs and symptoms vary depending in part on the type and severity of the CHD. Common signs include cyanosis, rapid breathing, rapid heartbeat, swelling in the hands, legs, ankle, feet and around the eyes, shortness of breath in babies during feeding (making it difficult for them to gain weight) and in older children, extreme tiredness and fatigue and fainting during exercise. There are several diagnostic tests for CHDs, including fetal and neonatal echocardiography, electrocardiogram, chest x-ray, pulse oximetry, and catheterization. Fetal echocardiography and in a relatively small proportion of cases, magnetic resonance imaging (MRI) are diagnostic methods for prenatal diagnosis of CHDs. As early as 16 weeks of gestational age, most major structural congenital heart defects can be detected. Currently, approximately 30% of all CHD cases are prenatally diagnosed in our population, with substantially higher proportions when minor defects are excluded [1,3,4]. The definitive diagnosis of a CHD is based on a specialized neonatal echocardiography and there may be a mismatch between the fetal and neonatal echocardiography findings.

Treatment of CHDs consists of a range of different strategies from medications to surgery depending on the severity and complexity of the malformation [4]. Currently, with the progress in trans-catheter and surgical techniques, newborns with complex CHDs have a substantially lower infant mortality and higher life expectancy than in the past. However, adverse neuro-developmental outcomes may occur in newborns with more severe forms of CHD. As a result, children with CHDs are more likely to miss school and to utilize additional healthcare, especially at younger ages [5,6].

CHDs are the most common group of congenital malformations with a total prevalence of about 8 per 1000 births [2,3,7,8]. The vast majority (>80%) of CHDs, and more so in the case of “isolated” CHDs, are live births, ~15% terminations of pregnancy for fetal anomaly (TOPFA), and 2% stillbirths (at >20 weeks of gestation). Maternal characteristics, particularly socioeconomic factors, can affect the probability of TOPFA [9].

The total prevalence of CHDs has increased over time, most likely due to improvements in diagnostic methods which help to detect minor cases that would previously go undetected. However, the live birth prevalence of severe CHDs (e.g., left ventricular outflow tract obstruction) has decreased due to prenatal diagnosis followed by TOPFA [5,7].

The prevalence of CHDs may vary across countries and regions. These differences may be related to differences in pre- and post-natal diagnosis, socioeconomic status of individuals living in particular geographical regions, as well as genetic and environmental factors. In low-income countries, lack of access to high-quality, specialized services may lead to under-estimation of the prevalence of CHDs [7].

CHDs have multifactorial causes, including genetic and epigenetic abnormalities as well as environmental factors in the broad sense of the term. The latter include a heterogeneous set of risk factors including maternal folic acid deficiency, rubella infection, gestational diabetes, alcohol abuse, and medications, including thalidomide and lithium. Air pollution might also be a risk factor for CHDs. While the estimates are not necessarily precise, approximately 20% of CHDs may be due to chromosomal anomalies and genetic disorders [5,9,10,11].

CHDs represent a wide spectrum of heterogeneous anomalies that show considerable variability in their prevalence, anatomy, developmental origin, severity, modalities of clinical and surgical management, short- and longer-term mortality, morbidity, and neuro-developmental outcomes. Given this heterogeneity, coding and classification of CHDs becomes a major and challenging question. Moreover, different classifications of CHDs may be needed to address different questions and hypotheses [11,12].

Currently, the most widely used coding and classification for CHDs is the 10th version of the International Classification of Disease (ICD10). Nevertheless, it is increasingly recognized, particularly by pediatric cardiologists and cardiac surgeons, that ICD10 has important shortcomings. In order to address these shortcomings, a comprehensive coding system, the International Pediatric and Congenital Cardiac Code (IPCCC) was devised. The IPCCC has many advantages but is complex and requires highly specialized coding. Moreover, given the number of codes in the IPCCC (the long list of IPCCC includes more than 10,000 individual codes), its use in clinical and epidemiological studies requires regrouping of individual anomalies [11].

In 2011, Houyel et al. [3,11] proposed the Anatomic and Clinical Classification of CHD (ACC-CHD) by rearranging the long list of the existing IPCCC into a manageable number of categories. The rearrangement is based on the cardiac anatomy and clinical features of the CHD. It consists of 10 main categories and 23 subcategories. It is intended to facilitate both the coding process and the analysis of the data in the setting of clinical and epidemiological studies. Timing of diagnosis, TOPFA, risk, and timing of infant mortality have been shown to be highly variable across the categories of CHDs in ACC-CHD [3]. This suggests that ACC-CHD may be a useful measure to predict the outcomes of CHDs.

The objectives of the study were to: (1) Describe the long-term childhood survival of newborns with CHD, and (2) Develop and validate predictive models of infant mortality based on the ACC-CHD.

## 2. Methods

### 2.1. Data Source

EPICARD (Epidémologie des Cardiopathies Congénitales) was a population-based, prospective, cohort study with long-term follow-up of children with a structural CHD born to women in the Greater Paris area (Paris and its surrounding suburbs). All cases (live births, terminations of pregnancy for fetal anomaly (TOPFA), fetal deaths ≥ 20 weeks) diagnosed in the prenatal period or up to 1 year of age in the birth cohorts between 1 May 2005 and 30 April 2008 were eligible for inclusion. Diagnoses were confirmed in specialized pediatric cardiology departments and for the majority of TOPFA and fetal deaths by a standardized pathology examination. When a pathology exam could not be performed, the diagnoses were confirmed by a pediatric cardiologist and a specialist in echocardiography, using the results of prenatal echocardiography examination. Multiple sources of data including all maternity units, pediatric cardiology and cardiac surgery centers, fetal and neonatal pathology departments, neonatal and pediatric intensive care units, infant units, and outpatient clinics were regularly consulted. Informed consent was obtained from parents for initial registration and from both parents and children at the 8-year follow-up. The study was approved by an ethics committee (French National Committee of information and Liberty, CNIL). Follow-up of children included assessment of the children’s health and neuro-developmental outcomes at 8 years of age [3].

### 2.2. Study Population

The total number of newborns included in the EPICARD study was 2867. After excluding TOPFA (N = 466) and fetal deaths (N = 53), our initial study population comprised 2348 live births. We excluded cases of CHDs associated with chromosomal (N = 149) or other anomalies (N = 318). The total study population was, hence, 1881 newborns with an “isolated” CHD.

### 2.3. Outcome and Predictor Variables

Our outcome variable was the survival to 8 years of life. The main predictor variable was the ACC-CHD. Due to the small sample sizes in the three categories 1, 5 and 10, these categories were combined, and so were categories 3 and 7, which had very few events (low mortality rate). Therefore, instead of the ten main categories of ACC-CHD, we used a simpler form of ACC-CHD with seven categories. The additional predictor variables comprised gender, gestational age (as a continuous variable modeled as a fractional polynomial), small for gestational age (birth weight below the 10th percentile in our population), as well as the number of cardiac surgeries during the first year of life.

We estimated two models: one with only the categories of ACC-CHD as predictor variables (model 1) and a second, more inclusive, model including ACC-CHD and additional predictor variables known to be related to the probability of survival in newborns in general (model 2). These additional variables were gender, small for gestational age, preterm delivery, and surgery during the 1st year of life.

### 2.4. Statistical Analysis

We used the Kaplan–Meier method for describing the survival rates. We constructed Kaplan–Meier plots for all newborns combined and for each category of the ACC-CHD. We used the Wilcoxon test to compare survival rates across the categories of ACC-CHD.

We used the Royston–Parmar flexible parametric approach to develop predictive models [13,14,15,16,17,18]. These models have certain advantages over both the Cox proportional hazards and parametric survival models. In contrast to the Cox model, these models can give estimations of the baseline hazards and provide smooth survival functions, thereby simplifying the interpretation of the plots without potential overemphasis on local features. They are also useful when the proportional hazards assumption is violated. The flexible parametric models also overcome a limitation of the standard parametric models in that they can more satisfactorily represent real data.

The flexible parametric models use three scales: hazard, odds, and probit. These are generalizations of the standard parametric Weibull, loglogistic, and lognormal models, respectively (Table 1). In standard parametric models, we assume that the effect of covariates is proportional on a given scale. We also assume that there is a linear relation between a particular transformation of the survival function and the logarithm of the survival time. In the Royston and Parmar model, this assumption is relaxed using restricted cubic splines, which allow a more robust estimation of the baseline survivor function [15].

The “dfs” is defined as the total number of knots (interior knots plus two boundary knots) minus one. The knots are chosen to be relatively close to the median uncensored log survival time in order to allow the data to be most closely modeled in the region of greatest density (Table 2) [15].

The choice of the scale and number of knots is made by comparing Akaike information criterion (AIC) or Bayes information criterion (BIC) statistics obtained from different scales with different degrees of freedom [15]. Royston and Parmar note that the optimal positioning of the knots is not critical for a good fit and may even be undesirable, in that the fitted curve may follow small-scale features of the data too closely. They also suggest using dfs of two or three for small datasets and five or six for larger datasets [15]. We found the probit scale and three degrees of freedom as the best parameters for our models.

### 2.5. Model Fit

We used martingale-like residuals to assess the goodness of fit of models for continuous variables. The martingale-like residual, r_i_, for the ith observation is r_i_ = δ_i_ − Hi^(t_i_), where δ_i_ is the censoring indicator and Hi^(t_i_) is the estimated cumulative hazard at the individual’s failure or censoring time, t_i_. If the model is correct, then E(r_i_|x_i_) = 0 for any x in the model and E(r_i_|x_i_$\hat{\beta }$) = 0 [15]. We also used the AIC and BIC to compare model fits. For a predictive model [13], the accuracy of predictions is the most important criterion to assess competing models. Indeed, all else being equal, a more accurate model that violates some assumptions may be preferred to a less accurate model that meets the relevant assumptions. However, in most cases, a model that meets the relevant assumptions tends to produce better predictions.

### 2.6. Assessment of Predictive Ability

We used calibration, discrimination, and a measure of explained variance for assessing the predictive ability of the models [13,14,15,16,17,18]. We used the Royston and Sauerbrei’s D statistic and $R_{D}^{2}$, which are measures of discrimination for survival models; D is a measure of prognostic separation of the survival curves—an estimate of the variance in the prognostic index of the model (xβ) across individuals—and $R_{D}^{2}$ is a measure of explained variation on the natural scale of the model. A simulation study showed that $R_{D}^{2}$ has better performance in comparison with the other measures of discrimination for survival models. It also has a more intuitive interpretation. Moreover, $R_{D}^{2}$ is robust in relation to outliers [19].

### 2.7. Model Validation

We used bootstrap for internal validation in order to obtain the corrected estimation of our performance index (R^2^). Firstly, we developed the model using all subjects and the performance index “apparent $R_{D}^{2}$.” was calculated ($R_{D\_app}^{2}$). Then, this measure was recalculated on a bootstrap sample with replacement ($R_{D\_boot}^{2}$) and the “optimism” in the fit from the bootstrap sample was calculated by $R_{D\_boot}^{2}$ − $R_{original}^{2}$; the optimism calculation was repeated 500 times. Finally, the bootstrap-corrected original performance was obtained by subtracting the average optimism from the apparent $R_{D}^{2}$ [20,21].
$$R_{D\_corrected}^{2}=R_{D\_app}^{2}-\frac{\sum {(R}_{D\_boot}^{2}-R_{original}^{2})}{n}\;$$

### 2.8. Comparison with the General Population

We compared the survival curves for each category of ACC-CHD with the population-level survival curves as provided by the INSEE (Institut National de la Statistique et des Etudes Economiques) from 2012 to 2016. Mortality rates were available for both genders and by different age groups [22].

### 2.9. Ethics Approval

The EPICARD study received ethics approval for data collection and anonymized data analyses by the French National Ethics Committee, the CNIL, on 13 March 2003 (approval no. 903006) before the study was launched. We also obtained authorization and ethics approval for continuing the long-term follow-up of children up to 8 years of age from both parents and the CNIL in 2008.

## 3. Results

### 3.1. Descriptive Analysis

Overall, among the 1881 newborns with isolated CHD, 1808 (96%) survived until eight years of age. We found no significant difference between survival rates by gender The descriptive analyses showed that newborns who were born at term, were not small for gestational age and those who did not have surgery during the first year of life had higher survival rates (Table 3).

More than 50% of the newborns belonged to the ACC-CHD group “anomalies of atria, interatrial communications and ventricular septal defects”. This group had the highest survival rate among the ten ACC-CHD groups. Newborns in the “ventriculo-arterial connections” and “anomalies of the extrapericardial arterial trunks” groups survived until 8 years. Newborns with “heterotaxy”, “complex anomalies of atrioventricular connections”, “coronary anomalies”, “anomalies of venous return”, and “anomalies of atrioventricular junctions and valves” had intermediate survival rates. We found the lowest survival rate in newborns with “functionally univentricular hearts”, with a survival rate of 44% at 8 years of age. The Wilcoxon test showed a significant difference between the survival rates across ACC-CHD categories (Table 4).

### 3.2. Predictive Models

We estimated two predictive models using the flexible parametric survival approach; one with the ACC-CHD categories as the only predictor variables and a second with the ACC-CHD categories plus other predictor variables (see Section 2). The predictive ability of the model that only included ACC-CHD categories was similar to the model that also included other variables.

We divided the predicted probabilities of survival into three categories (Figure 1). The category with the highest (>90%) survival probability included anomalies of the atria and interatrial communications and ventricular septal defects, ventriculo-arterial connections and anomalies of the extrapericardial arterial trunks. The categories with moderate (70–80%) survival probabilities included heterotaxy, AV connections, and coronary anomalies; anomalies of the venous return; and anomalies of the atrioventricular junctions and valves. The category with the lowest probability of survival (<50%) corresponded to the functionally univentricular hearts.

The instantaneous hazards peaked at the 3rd day of life for the category with the lowest survival probability, i.e., functionally univentricular hearts; this occurred at the 5th day of life for the category with moderate probabilities of survival and 8–11th day for the category with the highest probability of survival. The hazard rates at their peak were less than 1% and tended towards zero thereafter (Detailed information are available in Appendix A).

Table 5 shows the predictive performance of two models in the original dataset and bootstrapped samples. The D statistics were 1.43 and 1.60 for models 1 and 2, respectively. The higher D in model 2 shows that the variance in the prognostic index (xβ) across individuals is slightly greater in model 2. The $R_{D}^{2}$ of models 1 and 2 were 45% and 50%, respectively, in the original dataset. This implies that models 1 and 2 could explain 45% and 50% of the variation in survival rates among individuals. The corrected D statistics estimated in bootstrapped samples were 1.40 and 1.49 with $R_{D}^{2}$ of 0.44 and 0.47 for models 1 and 2, respectively. Hence, the predictive ability of models 1 and 2 on the original data and bootstrap samples were similar. 

## 4. Discussion

In this population-based, prospective cohort study of 1881 newborns with isolated CHDs (EPICARD study), we found that by far, the majority of newborns survived until 8 years of age. Survival rates varied greatly across different types of CHD, categorized based on an a priori, exhaustive classification of CHDs, the Anatomic and Clinical Classification of CHD (ACC-CHD). The lowest survival rate was for newborns with functionally univentricular hearts. Most deaths occurred in the first few days of life and hazard rates were quite low thereafter.

We developed and assessed the predictive ability of models for predicting the risk of mortality. We assessed both a model with ACC-CHD as the only predictor variable and a second model, which included additional variables known to be associated with the risk of mortality in newborns: gestational age, gender, small for gestational age, and cardiac surgery during the first year of life.

The predictive ability of the model with ACC-CHD alone was similar to the model that included additional variables. Therefore, after taking into account of the type of CHD with the ACC-CHD classification, other variables added little to the model’s predictive ability. The ACC-CHD model had rather good predictive ability based on the statistics proposed by Royston and Sauerbrei for measuring discrimination. Specifically, the model could account for about half of the differences in survival rates. This may be considered good to excellent discrimination as it corresponds to the higher end of the interval (0 to 60%) in most real data, as noted by Royston.

The probit-scale models provided the best fit for our data. However, these models are less familiar in the biomedical literature (even if widely used in econometrics) and their results are more difficult to communicate. As the “raw” results of probit models are less useful for clinicians, we provided instead the predicted probabilities of survival based on model predictions.

We found that that the survival rate of the children in our reference category, which included minor CHDs (mostly uncomplicated VSDS), was not significantly different from that of the general population. Hence, the estimated hazards for different categories of ACC-CHD represented the excess risk of mortality due to CHDs.

This is the first population-based study to assess the predictive ability of an exhaustive, a priori classification of structural CHDs for the long-term survival of newborns with CHD. The previous literature [22,23,24,25,26,27] has focused only on children who underwent surgical interventions. Moreover, most previous studies were based on data from specialized referral centers and assessed short-term, post-surgical (typically 30-days post-op) survival.

Our study has certain limits. Several of the ACC-CHD categories comprised a small number of newborns as they represented rare anomalies. Consequently, the survival measures were estimated with wide confidence intervals for some categories. It is worth noting, however, that our main goal was to develop and test the predictive ability of a model for predicting the overall survival of a group of newborns with CHDs. We did not aim to provide predictions for individual subgroups of CHDs. This was also the reason we combined small categories with similar survival rates.

Whereas the ten main categories of ACC-CHD clearly capture much of the heterogeneity in the risk of mortality associated with different types of CHDs, there remain important heterogeneities that are not explained by our model. Finally, additional studies are necessary for external validation of this model and to assess its potential impact for improving clinical decisions and practice evaluations.

## 5. Conclusions

The Anatomic and Clinical Classification of CHDs provided the basis for a model with good discrimination for predicting the long-term childhood survival of newborns with an isolated CHD. Prediction of survival for individual newborns with different types of CHDs remains an important challenge.

## Figures and Tables

**Figure 1 jcm-13-01623-f001:**
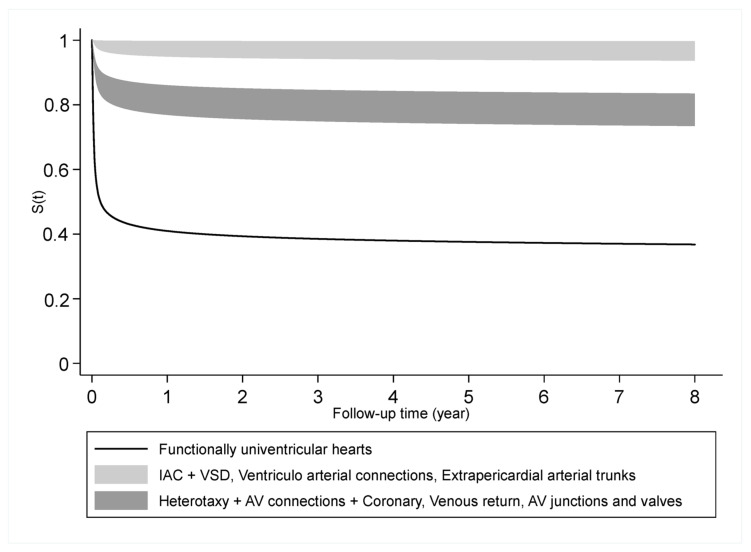
Predicted survival probabilities in different ACC-CHD groups estimated by model 2 with ACC-CHD and other predictors at their average.

**Table 1 jcm-13-01623-t001:** Flexible parametric survival different scales and their relative parametric models.

Standard Parametric Models	Flexible Parametric Scale
Weibull	Hazard
ln{−ln S(t; x_i_)} = ln{−ln S_0_(t)} + x_i_β	ln{−ln S_0_(t; x_i_)} = s(ln t; γ) + x_i_β
Loglogistic	Odds
logit{1 − S(t; x_i_)} = logit{1 − S_0_(t)} + x_i_β	logit{1 − S(t; x_i_)} = s(ln t; γ) + x_i_β
Lognormal	Probit
−Φ^−1^{S(ln t)} = −Φ^−1^{S_0_(ln t)} + x_i_β	−Φ^−1^{S(ln t)} = s(ln t; γ) + x_i_β

**Table 2 jcm-13-01623-t002:** Interior knots and degrees of freedom (dfs).

Number of Interior Knots	dfs	Centiles (Uncensored Log Event-Time)
1	2	50
2	3	33, 67
3	4	25, 50, 75
4	5	20, 40, 60, 80
5	6	17, 33, 50, 67, 83
6	7	14, 29, 43, 57, 71, 86
7	8	12.5, 25, 37.5, 50, 62.5, 75, 87.5
8	9	11.1, 22.2, 33.3, 44.4, 55.6, 66.7, 77.8, 88.9
9	10	10, 20, 30, 40, 50, 60, 70, 80, 90

**Table 3 jcm-13-01623-t003:** Descriptive characteristics of study population.

	N (%) or Mean ± SD	8-Year Survival Rate (%)
Female	1018 (54.1)	96
SGA ^†^ (<10th percentile of Audipog curve)	205 (10.9)	93 *
Surgery during 1st year of life	382 (20.3)	92 *
Preterm (<37 weeks of gestation)	240 (12.8)	91 *

* *p* < 0.05, logrank test; ^†^ small for gestational age.

**Table 4 jcm-13-01623-t004:** Distribution of ACC-CHD groups and their survival rates.

ACC-CHD Groups	N (%)	S(t) *
Heterotaxy, including isomerism and mirror-imagery + complex anomalies of atrioventricular connections + congenital anomalies of the coronary arteries (heterotaxy + AV connections + coronary anomalies)	22 (1.2)	0.68 [0.45–0.83]
Anomalies of the venous return	20 (1.1)	0.80 [0.55–0.92]
Anomalies of the atria and interatrial communications + ventricular septal defects (IAC + VSD)	1311 (69.7)	0.99 [0.988–0.997]
Anomalies of the atrioventricular junctions and valves	48 (2.5)	0.77 [0.62–0.87]
Functionally univentricular hearts	34 (1.8)	0.44 [0.27–0.60]
Anomalies of the ventricular outflow tracts (ventriculo-arterialconnections)	360 (19.1)	0.94 [0.91–0.96]
Anomalies of the extrapericardial arterial trunks	86 (4.6)	0.92 [0.84–0.96]

* 8-year survival rate and its 95% confidence interval in brackets.

**Table 5 jcm-13-01623-t005:** Predictive ability of model 1 (ACC-CHD as the only predictor) and the full model (model 2) with ACC-CHD and other predictors.

	Model 1	Model 2
Original dataset		
Royston’s D	1.43	1.60
R^2^	0.45 [0.36–0.52]	0.50 [0.41–0.58]
Bootstrap sampling		
Royston’s D	1.40	1.49
R^2^	0.44 [0.35–0.51]	0.47 [0.38–0.55]

## Data Availability

The data presented in this study are available on request from the corresponding author if needed for a collaborative study. The data are not publicly available due to confidentiality reasons.

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
