# Peer review of "Predicting Long-Term Childhood Survival of Newborns with Congenital Heart Defects: A Population-Based, Prospective Cohort Study (EPICARD)"

_jcm, 2024, doi:10.3390/jcm13061623_

Round 1

Reviewer 1 Report

Comments and Suggestions for Authors

This is an interesting paper, even if the sample is limited, addressing some important topics regarding congenital heart diseases, among which a simplified and practical classification and an approach to the prediction of long-term survival. The work may be improved with a more concise, focused, and synthetic introduction, a reduction of the tables and figures, and the discussion and conclusions should be separated.

Comments on the Quality of English Language

The language requires some editing for a clearer presentation of the study.

Author Response

We thank our reviewer for positive the comments and constructive criticisms. In the revised version submitted, we have done extensive editing of the manuscript to streamline particularly the Introduction and Discussion sections and separated the Conclusions from Discussion as requested. We defer to the Editor as to which Tables and Figures should be included in the main manuscript and which ones  included in an accompanying appendix.

Reviewer 2 Report

Comments and Suggestions for Authors

The paper is technically sound. The data support the conclusions. However, it needs revision:

1.     Title: good

2.     Abstract: All abbreviations need revision, please remove all abbreviations from abstract as possible.

3.     Introduction: Please add data about  the International Pediatric and Congenital Cardiac Code (IPCCC).

4.     The study methodology: 

a.     The Materials and Method sections are good, however the exclusion criteria should be clearer.

b.     Part 2.7 is missing and appeared as a picture with missing parts please edit.

5.     Results: very good. But figure2 is not clear and need a new one with clear data.

6.     Discussion: very good.

                   7.  The conclusion contains a suggestion for future work, please conclude your findings only and transfer the suggestions to the end of the discussion

Comments on the Quality of English Language

 Minor editing of English language required

Author Response

We thank our reviewer for the careful reading of our manuscript and constructive comments and critcisms. In the revised version we have modified the manuscript as follows in order to address our reviewer's recommendations:

Abstract has been revised including removal of any acronyms that are not explicitly explained in the text.

We think the revised version makes the exclusion criteria sufficiently clear as a clarification was requested by our reviewer.

Section 2.7 was modified in the process of formatting by JCM. We have written to the Editors about this point and have provided the corresponding section in the correct format from our originally submitted manuscript.

We have removed the future research section from the conclusion and moved it to the Discussion section as suggested by our reviewer.

About Figure 2, we are not clear about our reviewer's comment. The figure correctly reflects our data but is based on predictions from the model. In any case, since the hazard figure (Fig. 2) is the "inverse" / mirror image of the survival curve, it can be deleted. Since there was also a comment by another reviewer about the number of figures, Figure 2 can be either deleted or included in an appendix. We defer to our reviewers / editors as to which option is better. 

Reviewer 3 Report

Comments and Suggestions for Authors

I have read with interest the manuscript by Rahshenas et al. “Predicting long-term childhood survival of newborns with congenital heart defects: A population-based, prospective cohort study (EPICARD)”

Among the remarks and commentaries:

As the authors correctly states, most of the current literature which focuses on long-term outcome in patients with CHD is based on institutional, multi-centric or registry data of surgical patients or interventional cardiology patients.

This report is original as it includes all “incomers” as I understood.

Unfortunately, the introduction is far too long and should be thoroughly revised

As a general comment, the results do not add much to the current knowledge, confirming that the less severe the diagnosis of CHD (ACC-CHD classification, STAT, STS-EACTS risk categories).

In addition, even though a cohort of >1800 patients might appear sufficient, many of the ACC-CHD categories included less than 50 patients in the current study, as reflected by the wide confidence interval in the results provided.

Could the authors provide details on

a) the completeness of the follow-up

b) the median follow-up time

As the inclusion ended in 2008, and a complete follow-up, a median time greater than 15 years is expected.

Why the authors did choose the outcome variable to be survival up to 8 years of age? Ethical approval did not allow to extend this f-up observational period?

Long-term outcome extend beyond 10-15 years, my understanding is that parents and relatives to those pediatric patients would be eager to have informations that extend over 8 years of age… even though the data demonstrate that most deaths are encountered within the 2-3 years of life.

c) how many cardiac CHD centres included patients in the prospectively enrolled patients? And what was the proportion of “conservatively” treated patients relative to those who benefited from ANY intervention (percutaneous and/or surgical) ?

Among the details, figures and legends (notation of axis) must be improved.

The discussion is weak as the authors mostly underscore that the simplification to the statistical model does not jeopardize the results.

Refs R + A +STS should be clarified to the readers.

Author Response

We thank our reviewer for the careful reading of our manuscript and for the positive comments and constructive criticisms. Below we have provided a reply to each comment and noted changes made in the revised manuscript as relevant.

We agree with our reviewer about sample size issues. We have acknowledged this point, noting in the Discussion that despite the large size of the cohort, several ACC-CHD categories were small in number and others had high numbers but low mortality and hence a relatively low number of events. However, the main objective and result here is not the mortality associated with any given category of CHD; instead the main criteria of interest are those that measure the predictive ability of the overall ACC-CHD classification for predicting childhood mortality for newborns with CHD. We have noted this point in the Discussion.

We also agree with our reviewer that the main contribution of this paper is not descriptive but rather the predictive modelling. However, for readers not familiar with the cohort and our previous work, it seemed more appropriate to include at least briefly the descriptive results associated with each ACC-CHD group. In addition, to our knowledge, this is the first time that both the probablity and timing of long-term childhood mortality is reported in a population-based prospective cohort. In our previous descriptive work, we had only reported on the probability of death and not its timing. So we thought that purely descriptive results are worth reporting as well.

About the length of follow-up and why we report only on 8-year mortality (and not on a longer period given the age of the children in the cohort), a few points are worth mentioning. The last inclusions were in 2009 and the follow-up took place between 2015-2017. We did not follow the children afterwards as our funding was for a 8-year period. The main outcomes of interest were those related to neuro-developmental outcomes of survivors, which was time-consuming and costly to obtain, and that we have published previously. Mortality was also of interest as population-based cohort data on a large number of newborns with CHD are rare in the literature. Moreover, our development and validation of a prediction model for all CHD has not been done before so we felt it was a worthwhile study. 

We have added the references on surgical severity scores that were omitted because of a formatting error.

About the treatment strategies (conservative vs. interventionist) in different centers, we did not have data on relevant parameters and our clinician colleagues noted that in the same center therapeutic decisions are not always consistent and may depend on the attending in charge of the patient. In any case, our objective was to develope a predictive model that reflects the overall practices and outcomes in our population.  

Round 2

Reviewer 2 Report

Comments and Suggestions for Authors

The authors responded to all issues.

Author Response

We thank our reviewer for the careful reading of our manuscript and constructive comments. We are glad that we were able to address the questions and comments satisfactorily.